# Incremental Bag of Words with Gradient Orientation Histogram for Appearance-Based Loop Closure Detection

**Yuni Li** [1], **Wu Wei** [1,*] and **Honglei Zhu** [2]

1   College of Automation Science and Engineering, South China University of Technology, Guangzhou 510641, China
2   Guangzhou Institute of Technology, Guangzhou 510075, China; hongxue0571@126.com
*   Correspondence: weiwu@scut.edu.cn

**Abstract:** This paper proposes a novel approach for appearance-based loop closure detection using incremental Bag of Words (BoW) with gradient orientation histograms. The presented approach involves dividing and clustering image blocks into local region features and representing them using gradient orientation histograms. To improve the efficiency of the loop closure detection process, the vocabulary Clustering Feature (CF) tree is generated and updated in real time using the Balanced Iterative Reducing and Clustering using Hierarchies (BIRCH) algorithm, which is combined with an inverted index for the efficient selection of candidates and calculation of similarity. Moreover, temporally close and highly similar images are grouped to generate islands, which enhances the accuracy and efficiency of the loop closure detection process. The proposed approach is evaluated on publicly available datasets, and the results demonstrate high recall and precision.

**Keywords:** simultaneous localization and mapping (SLAM); loop closure detection; Bag of Words

## 1. Introduction

In the field of autonomous robotics, loop closure detection plays a crucial role in simultaneous localization and mapping (SLAM) systems. As the SLAM system accumulates errors during long exploratory periods, constraints need to be generated when areas are re-observed in order to achieve consistent maps. As such, a robust loop closure detection mechanism is vital for accurate localization and mapping results. Appearance-based loop closure detection algorithms [1–6] have gained increasing attention due to the rapid development of visual SLAM algorithms that use cameras as external perception sensors.

The performance of an appearance-based loop closure detection algorithm is highly dependent on the image description method and the ability to retrieve similar images from previous observations. Image description can be categorized into local feature descriptors [7–10] and global feature descriptors [11–13]. Global feature descriptors are less sensitive to changes in lighting conditions, while local feature descriptors are better adapted to changes in viewpoint. The BoW model [14] has been proven to be an effective solution for image indexing, but it typically requires offline training to generate the visual vocabulary, which can be time-consuming and its applicable environment limited by the training set. An alternative approach is to construct the vocabulary from the current environment in an online and incremental manner.

In this paper, we present a novel approach for appearance-based loop closure detection using gradient orientation histograms as local region feature descriptors, and an online, incremental method for building the vocabulary based on these features. To address the problems that the extraction of the local feature descriptor is time-consuming, the local feature descriptor is sensitive to local changes, and the global descriptor performs poorly in scenes with large pose changes, a method for feature extraction using global descriptors for image regions instead of keypoints is proposed. This method divides image blocks and clusters them based on their gradient direction histograms to extract local region features,

which exhibit pose invariance while preserving the overall structure of the image region. Aiming at the problem that the traditional BoW model requires offline training and the effect of dictionary construction is easily affected by the training dataset, an incremental BoW model is proposed. To generate and update the vocabulary in real time, we use the BIRCH algorithm [15], which allows an incremental construction of the vocabulary CF tree and avoids the disadvantages of the offline method. The vocabulary CF tree in combination with an inverted index is used for the efficient selection of candidates and calculation of similarity. Finally, to avoid images taken from the same location competing among themselves as loop closure candidates, we group temporally close and highly similar images to generate islands.

The rest of this paper is organized as follows: Section 2 provides a review of related work on appearance-based loop closure detection algorithms. Section 3 presents our new loop closure detection algorithm. In Section 4, we evaluate the performance of our algorithm on publicly available datasets. Finally, a brief summary is provided in Section 5.

## 2. Related Work

According to the method used to describe the input images, appearance-based loop closure detection falls into two broad categories: those that extract local feature descriptors from parts of the image; and those that use the whole image to compute global descriptors [16].

Global descriptors used in loop closure detection included color histograms [11], Gist descriptors [12] and descriptors based on principal component analysis [13]. Badino et al. used descriptors containing gradient information of the entire image known as WI-SURF to perform localization [17], and Sünderhauf et al. used BRIEF [9] as a holistic descriptor for a complete image [18]. Glocker et al. proposed an efficient keyframe-based relocalization method based on frame encoding using randomized ferns [19]. Arandjelovic et al. introduced a CNN architecture that is trainable in an end-to-end manner directly for the place recognition task [20]. Global descriptor methods are generally faster to compute and more robust to local appearance changes, but they can be less tolerant to changes in the robot's pose.

Local feature descriptor methods typically involve extracting distinctive regions or keypoints from images [21], and representing them with descriptors that encode their appearance or local geometry. Image similarity is then measured by comparing these descriptors between images. Since each image may contain a large number of local features, the BoW model [14] is often used to improve retrieval efficiency. The BoW model involves clustering local features into visual words, and uses the visual vocabulary to convert an image into a sparse numerical vector. A very successful algorithm, FAB-MAP 2.0 [2], which employs a vocabulary of up to 100,000 visual words and approximated the probabilities of visual word co-occurrences using a Chow–Liu tree. Gálvez-López et al. built a vocabulary tree based on BRIEF [9] to discretize a binary descriptor space and use the tree to speed up correspondences for geometrical verification [4]. Building upon this work, Mur-Artal et al. proposed a BoW place recognizer with ORB [10] features to achieve higher robustness and faster speed [22]. Most of the related work mentioned above relies on predefined vocabularies that are constructed from features extracted from a training image sequence. To address the limitations of this approach, Nicosevici et al. [23] proposed an online method for continuously updating the vocabulary based on observations, while still maintaining the ability to match prior observations with future ones. Additionally, Khan et al. [5] introduced an incremental BoW approach that builds a vocabulary consisting of binary visual words in an online, incremental manner by tracking features between consecutive frames. Building on this algorithm [5], Garcia-Fidalgo et al. [6] used a hierarchical and incremental structure to reduce the complexity of the BoW assignment process.

To incorporate the advantages of both local and global descriptors, several loop closure detection methods using both techniques can be found in [24,25]. McManus et al. [24] computed the global descriptor HOG [26] on image patches to learn distinctive visual

elements called scene signatures. Sünderhauf et al. [25] used a convolutional network (ConvNet) to extract features for each region landmarks.

## 3. Proposed System

The pipeline of operations performed in the loop closure detection mechanism discussed in this paper is shown in Figure 1. The pipeline is composed of three parts: (i) feature extraction, (ii) the incremental BoW model, and (iii) the loop closure detection stage.

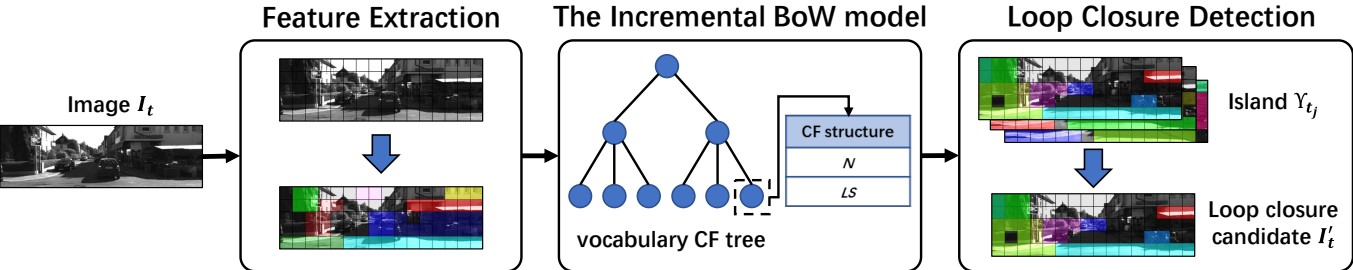

**Figure 1.** The pipeline of the loop closure detection mechanism.

### 3.1. Gradient Orientation Histogram Feature Extraction

The gradient orientation histogram is a global descriptor commonly used in image analysis and recognition, which can provide information about the appearance and shape of an image by the distribution of local intensity gradients. We divided the image into an $m \times n$ grid of sub-images, each no larger than the maximum pixel size. For each image block, the gradient orientation was quantified into 9 bins and weighted by the magnitude of the gradient.

After performing the operations mentioned above, we obtain 9-element vectors to represent image blocks, which are then clustered using a density-based spatial clustering algorithm. This clustering algorithm is inspired by DBSCAN (Density-Based Spatial Clustering of Applications with Noise) [27], which groups data points based on their proximity and density. An appropriate similarity measure is the cosine distance, which measures the similarity between two vectors by computing the cosine of the angle between them. The distance between two vectors $A$ and $B$, denoted as $d(A, B)$, is calculated as:

$$d(A, B) = 1 - \frac{A \cdot B}{\|A\| \times \|B\|},$$

(1)

this distance is a normalized measure with range [0, 1], which decreases as the similarity between the two vectors increases. The clustering algorithm can be divided into the following steps:

- Parameter Selection: Choose the core distance, $\epsilon_c$, and the reachability distance, $\epsilon_r$.
- Core Object Identification: Identify objects whose distance to at least one of its neighboring objects (image blocks adjacent this block upper, lower, left and right) is less than $\epsilon_c$. These objects are known as core objects.
- Reachable Object Identification: If the distance between an object and one of its neighboring objects is less than $\epsilon_r$, the object is considered a reachable object of this neighboring object.
- Cluster Generation: Assign each core object and its reachable objects to a cluster. For each reachable core object, add its reachable objects to the cluster until no new objects can be added.
- Output: Return the clusters.

An example of clustering image blocks using this algorithm is shown in Figure 2. By clustering similar image blocks, we extracted several image region features from the image. Each feature vector is obtained by summing the vectors of the image blocks in the region and normalizing the resulting vector to unit length.

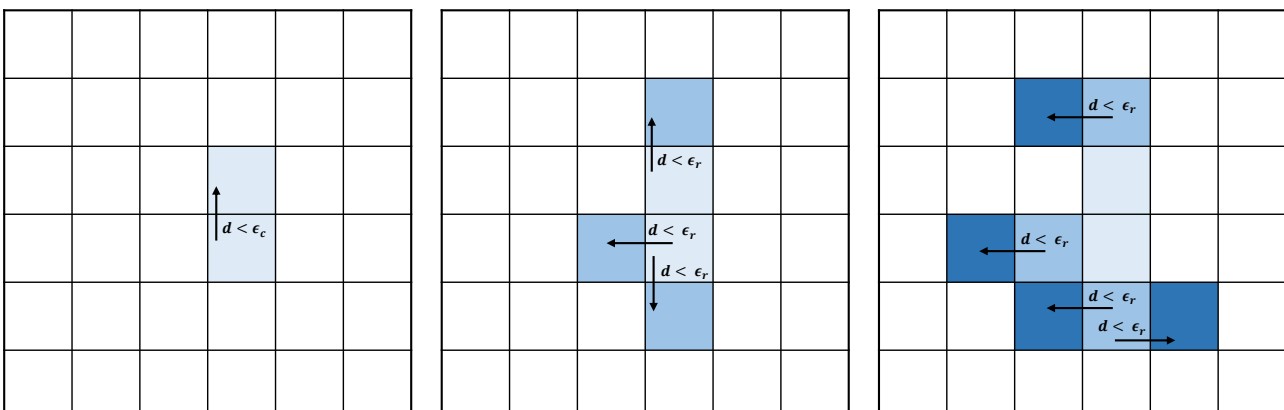

**Figure 2.** An example of clustering image blocks. Starting from a core object to look for its reachable objects. For each reachable object, add its reachable objects until no new objects can be added.

### 3.2. The Incremental BoW Model

In most hierarchical BoW approaches, a common method to build the vocabulary tree is to extract a rich set of features from some training images and perform k-medians clustering using *k*-means++ seeding [28]. In our proposed approach, we employ the BIRCH algorithm and update the vocabulary tree in a rapid and incremental manner.

BIRCH is a hierarchical clustering algorithm that is particularly suitable for large datasets, as it can efficiently cluster data points by incrementally processing the data objects and building a CF tree. The CF is a data structure used in BIRCH, which contains three attributes: $N$ (the number of data objects in the cluster represented by the CF), $LS$ (the linear sum of the data objects in the cluster represented by the CF), and $SS$ (the squared sum of the data objects in the cluster represented by the CF). The CF structure has the nice property that its attributes satisfy a linear relationship, i.e., the attributes of the CF are the sum of the attributes of its children.

In the vocabulary CF tree, the internal nodes represent clusters and the leaves represent words of the vocabulary. Each node and leaf contains a CF structure, where $N$ represents the sum of the number of image blocks of features in the CF, $LS$ represents the sum of the product of the feature vectors and the number of image blocks, and $SS$ is not used (since all feature vectors are unit vectors). When a new image region feature is extracted, it is added to the vocabulary CF tree by selecting the intermediate node that minimizes the distance at each level from root to leaf. If the addition of a new feature would cause a leaf to exceed the maximum radius of its hypersphere, the leaf is split into two leaves. Similarly, if the increase in the number of leaves would cause a node to exceed its maximum capacity, the node is split into two child nodes. Finally, all CF structures on the path are updated.

In addition to the vocabulary tree, an inverse index is also maintained and updated, which stores a list of images where each word in the vocabulary has appeared. By using the inverse index, we can speed up the generation of loop closure hypotheses, as we only need to compare images that have at least one word in common with the query image.

### 3.3. Loop Closure Detection

After the feature regions have been extracted from a new image and used to update the vocabulary CF tree, the loop closure detection phase can begin. First, a set of matching candidates for the new image can be obtained from the inverse index. Then, these images are compared and grouped for similarity based on the weight computed from the vocabulary CF tree. Finally, the best loop closure candidate is selected based on the similarity score.

#### 3.3.1. Searching for Matching Candidates

When an image, $I_t$, captured at time index $t$, is acquired, a list of images that have common words with the query image $I_t$ can be generated using the inverse index. For each image in the list, it is necessary to compare its region features with those of the query

image and remove the mismatched region feature pairs. When two images do not have a significant viewpoint change, their matching image regions generally have the same relative position in the image, and two images usually have only a few feature matching pairs (rather than tens or hundreds in the feature point method). Therefore, by exhaustively computing the average center of mass of these image regions, we can calculate the average relative positions of all matching pairs and remove any outliers.

Finally, we exclude images that do not reach the required number of matching features, resulting in a final list of matching images of image $I_t$, denoted as $I_{t_1}, I_{t_2}, \ldots$. These steps are crucial in accurately identifying similar images and ensuring reliable results.

### 3.3.2. Similarity Measure

To measure the similarity between images, it is necessary to convert the image into a BoW vector. To give weight to each word, we use the term frequency–inverse document frequency (*tf–idf*) [29], which is the product of the term frequency (*tf*) and the inverted document frequency (*idf*). For the image $I_t$, which contains words $w_1, \ldots, w_m$, the CF attributes can be used to efficiently calculate the $tf_i$ and the $idf_i$ of each word $w_i$ as follows:

$$tf_i = \frac{n_{w_i}}{\sum\limits_{j=1}^{m} n_{w_j}}. \tag{2}$$

$$idf_i = log \frac{N_{root}}{N_{w_i}}, \tag{3}$$

Here, $N_{w_i}$ represents the $N$ attribute of the CF structure of word $w_i$, $N_{root}$ denotes the $N$ attribute of the CF structure of the root node, and $n_{w_i}$ indicates the number of image blocks in the image region corresponding to the word $w_i$ in image $I_t$. By combining the $idf_i$ and $tf_i$ values as the weight of each word $w_i$, we can construct the BoW vector $v_t$ for the image $I_t$. This BoW vector is an efficient representation of the image that can be used for similarity comparison with other images.

Once the list of matching images $I_{t_1}, I_{t_2}, \ldots$, is obtained, they are converted into BoW vectors $v_{t_1}, v_{t_2}, \ldots$, and the similarity with $v_t$ is computed in turn. The similarity between two vectors $v_a$ and $v_b$ is measured by calculating the *L1-scores* $s(v_a, v_b)$:

$$s(v_a, v_b) = 1 - \frac{1}{2} \left| \frac{v_a}{|v_a|} - \frac{v_b}{|v_b|} \right|. \tag{4}$$

These scores are then normalized with the score of $v_t$ and $v_{t-1}$, where $v_{t-1}$ is the BoW vector of the previous image. The normalized similarity score $\widetilde{s}(v_t, v_{t_j})$ between $v_t$ and $v_{t_j}$ is calculated as

$$\widetilde{s}(v_t, v_{t_j}) = \frac{s(v_t, v_{t_j})}{s(v_t, v_{t-1})}, \tag{5}$$

The matches whose normalized similarity score $\widetilde{s}(v_t, v_{t_j})$ does not achieve a minimum threshold are rejected.

### 3.3.3. Islands Computation

To avoid images competing with each other, we refer to the method used in [4] and group those that are taken close in time into islands. An island, denoted as $Y_{t_i}$, is defined as a set of matching images whose time index are between $t_{n_i}$ and $t_{m_i}$, and the interval between consecutive time index is less than a certain threshold. Figure 3 illustrates the concept through a simple example.

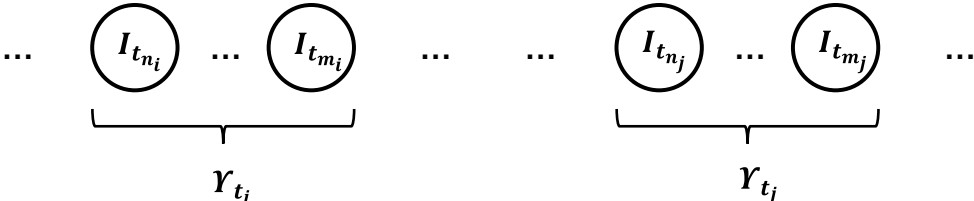

**Figure 3.** A simple example of islands. Island $Y_{t_i}$ contains images with the time index between $t_{n_i}$ and $t_{m_i}$, and island $Y_{t_j}$ contains images with the time index between $t_{n_j}$ and $t_{m_j}$.

To rank the islands, we compute a score $H$ for each of them. The score is computed using the function $H(v_t, V_{t_i})$:

$$H(v_t, V_{t_i}) = \sum_{j=n_i}^{m_i} \widetilde{s}(v_t, v_{t_j}), \tag{6}$$

where $V_{t_i}$ represents the set of all BoW vectors in the island $Y_{t_i}$.

After selecting the island $Y_{t_j}$ with the highest score $H(v_t, V_{t_j})$ as the matching group, the loop closure detection process continues to the temporal consistency step. Grouping similar images into islands helps to narrow down the search space and focus on finding the best match within a smaller subset, which improves the accuracy and efficiency of the loop closure detection process.

### 3.3.4. Temporal Consistency

After obtaining the best match island $Y_{t_j}$, a simple temporal consistency test is performed. This test is based on previously observed matching islands, and for the current image $I_t$, its best matching island $Y_{t_j}$ must have a significant overlap with the best matching islands detected in the time index interval of $t$ and $t - k$. A simple example of the temporal consistency verification process is shown in the Figure 4.

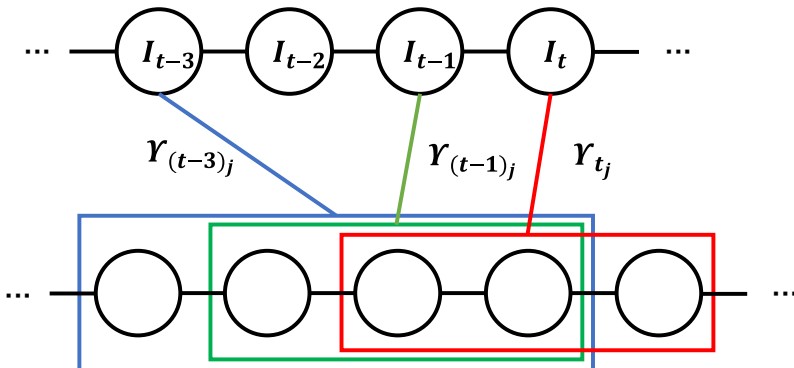

**Figure 4.** A simple example of the temporal consistency verification process. In the time index interval of $t$ and $t - k$, three images whose time indexes are $t$, $t - 1$, and $t - 3$ have basically consistent matching islands $Y_{t_j}$, $Y_{(t-1)_j}$, and $Y_{(t-3)_j}$, so $Y_{t_j}$ passes the time consistency verification.

If an island $Y_{t_j}$ passes the temporal constraint, the image $I'_t$ with the highest score $\widetilde{s}(v_t, v'_t)$ in $Y_{t_j}$ is considered a real loop closure. The temporal consistency test helps to reduce false positives and improve the accuracy of loop closure detection.

## 4. Experimental Results

In this section, we present an evaluation of our proposed algorithm. First, we outline the methodology we used to evaluate our algorithm. Next, we compare the results of feature extraction for different system parameters. Following this, we provide examples of correct and incorrect results of image matching in loop closure detection, along with the

associated similarity scores. Finally, we evaluate the vocabulary size and the performance of our system.

### 4.1. Methodology

The loop closure detection stage, a critical component of visual SLAM algorithms, typically relies on image data obtained from a visual SLAM front-end, which often involves image pre-processing and keyframe selection. In order to make a better fit for mainstream visual SLAM algorithms, we used DSO [30] as the front-end visual odometry to obtain photometrically calibrated and representative keyframe data for evaluating loop closure detection algorithms. DSO is a typical direct visual odometry approach that operates on the raw intensity values of image pixels, without relying on feature point extraction and matching.

In the feature extraction stage of our proposed method, the maximum pixel size of each image block is set to 80, the core distance $\epsilon_c$ is set to 0.02, and the reachability distance $\epsilon_r$ is set to 0.05. In the incremental BoW model, the maximum radius of a leaf is set to 0.005, and the maximum capacity of a node is set to 10. In the loop closure detection stage, the minimum normalized similarity score is set to 1.2, and the minimum consecutive time index interval is set to 5.

We evaluated our algorithm using precision and recall metrics, with precision being defined as the ratio of the number of correct detections to the total number of detections, and recall being defined as the ratio of the number of correct detections to all real loop closure events.

### 4.2. Dataset

We evaluated our system using the publicly available KITTI Odometry dataset [31]. The KITTI dataset is a widely used public dataset for evaluating visual SLAM algorithms in outdoor car driving scenarios. It encompasses diverse environments, including urban, rural, and highway scenes, captured by a vehicle equipped with multiple sensors such as several stereo cameras, GPS, lidar, and Inertial Measurement Unit (IMU). This paper utilized the visual odometry data from the KITTI dataset, which comprises extensive outdoor images of large-scale scenes. There are moving vehicles and pedestrians in some images, as well as various degrees of occlusion, and changes in light and weather conditions.

The KITTI visual odometry dataset consists of 22 image sequences, with 11 of them (KITTI 00–KITTI 10) accompanied by ground truth trajectories, allowing for the evaluation of SLAM systems' trajectory estimation. The image data include calibration files, color images from the left and right cameras, as well as grayscale images from the left and right cameras. In this paper, four sequences, namely KITTI 00, KITTI 05, KITTI 06, and KITTI 07, which containing loop closures, were selected for testing purposes. Only the sequence of left camera grayscale images was used to evaluate the monocular direct method visual SLAM system. The specific parameters associated with the four sequences (KITTI 00, KITTI 05, KITTI 06, and KITTI 07) are presented in Table 1.

To evaluate the performance of the loop closure detection method, we manually created a list of real loop closures, with each time interval in the list having a matched time interval.

**Table 1.** Parameters of KITTI visual odometry dataset.

| Dataset | Number of Images | Image Size (Height × Width) |
| --- | --- | --- |
| KITTI 00 | 4541 | 376 × 1241 |
| KITTI 05 | 2761 | 376 × 1241 |
| KITTI 06 | 1101 | 376 × 1241 |
| KITTI 07 | 1101 | 376 × 1241 |

### 4.3. Feature Extraction

In the process of feature extraction, the size of the image block is a critical factor that affects performance. Calculating the gradient orientation histogram for a larger image block can capture the overall structure of the block, but at the cost of losing more local details. On the other hand, calculating the gradient orientation histogram for a smaller image block can capture more details but is also too sensitive to deformations. We segment the image into different block sizes by setting the maximum pixel size of the image block. Figure 5 illustrates the feature extraction result of different maximum pixel size settings on an image from the KITTI07 dataset, where each local region feature is indicated by a different color. It can be observed that larger maximum pixel sizes result in fewer image blocks for clustering, leading to a loss of more image information. Conversely, smaller maximum pixel sizes increase the amount of less important information and significantly increase execution time.

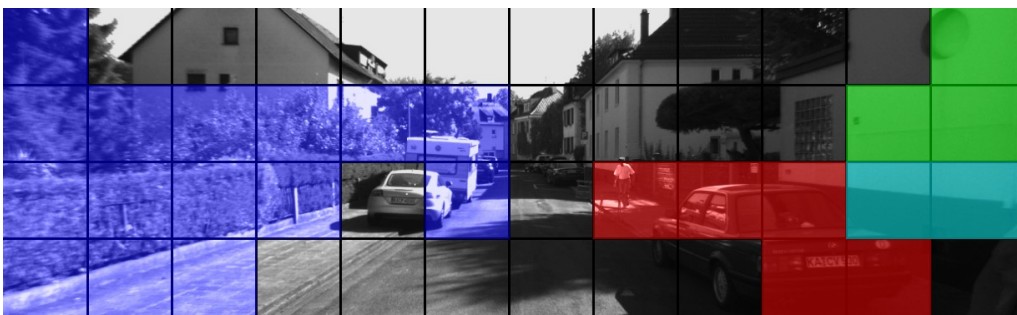

Maximum pixel size = 110

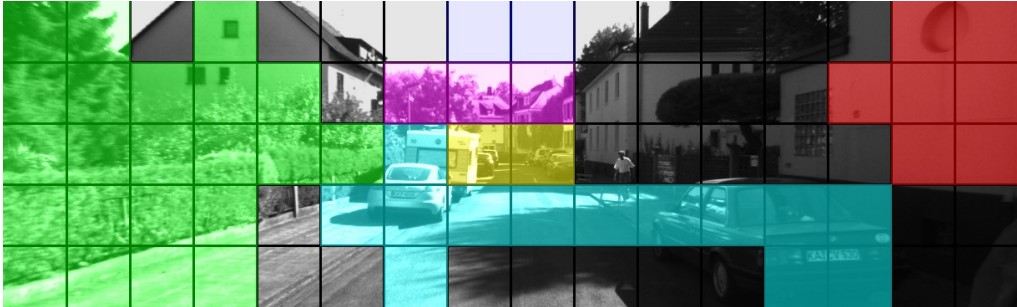

Maximum pixel size = 80

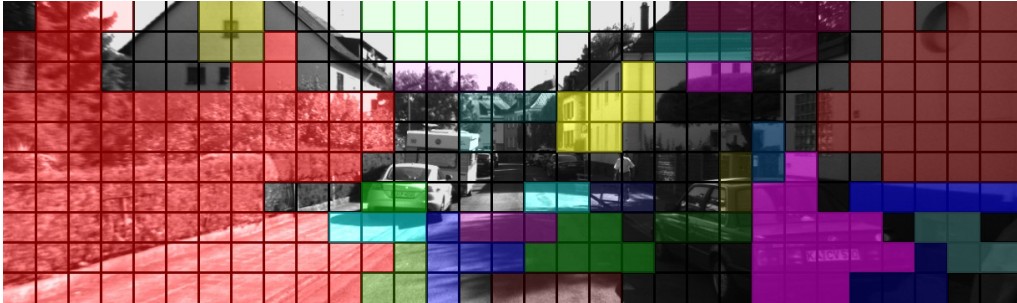

Maximum pixel size = 40

**Figure 5.** The feature extraction result of different maximum pixel size settings on an image from the KITTI07 dataset, where each local region feature is indicated by a different color.

After segmenting the image into image blocks, the core distance $\epsilon_c$ and the reachability distance $\epsilon_r$ are the two most critical parameters for clustering the image blocks. The $\epsilon_c$ determines the beginning of a cluster, while $\epsilon_r$ affects the size of the cluster, and the two parameters work together to generate a cluster. Figure 6 demonstrates the feature

extraction result of different $\epsilon_c$ and $\epsilon_r$ settings on the image shown in Figure 5. It is important to note that $\epsilon_r$ cannot be smaller than $\epsilon_c$. From Figure 6, we can observe that when $\epsilon_c$ is small, the number of clusters is small, while a large $\epsilon_c$ leads to an increase in the number of clusters, but some less important information is also extracted as a cluster. When $\epsilon_r$ is small, the size of the clusters is relatively small, and a larger $\epsilon_r$ results in the merging of adjacent clusters, which ultimately reduces the number of clusters obtained.

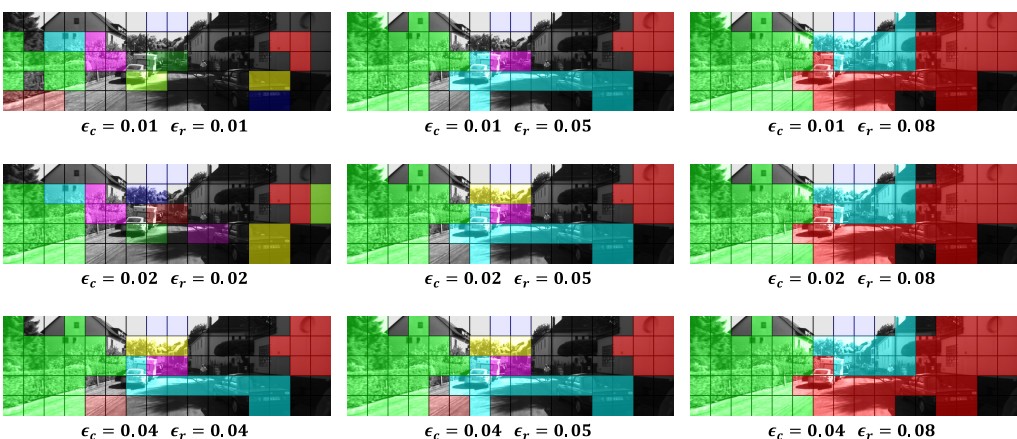

$\epsilon_c = 0.01 \quad \epsilon_r = 0.01$        $\epsilon_c = 0.01 \quad \epsilon_r = 0.05$        $\epsilon_c = 0.01 \quad \epsilon_r = 0.08$

$\epsilon_c = 0.02 \quad \epsilon_r = 0.02$        $\epsilon_c = 0.02 \quad \epsilon_r = 0.05$        $\epsilon_c = 0.02 \quad \epsilon_r = 0.08$

$\epsilon_c = 0.04 \quad \epsilon_r = 0.04$        $\epsilon_c = 0.04 \quad \epsilon_r = 0.05$        $\epsilon_c = 0.04 \quad \epsilon_r = 0.08$

**Figure 6.** The feature extraction result of different $\epsilon_c$ and $\epsilon_r$ settings on the image shown in Figure 5, where each local region feature is indicated by a different color.

### 4.4. Feature Matching

During the loop detection algorithm, the vocabulary CF tree is constantly updated, providing information for feature matching and similarity calculation between images. Figure 7 illustrates two sets of matching image results in the KITTI07 dataset. Both sets of matching images share common words, but the top result is the correct match while the bottom one is incorrect. By calculating the similarity through the vocabulary CF tree, the similarity score of the correct match is 0.391704, while the similarity score of the incorrect match is 0.0849734, showing a significant difference.

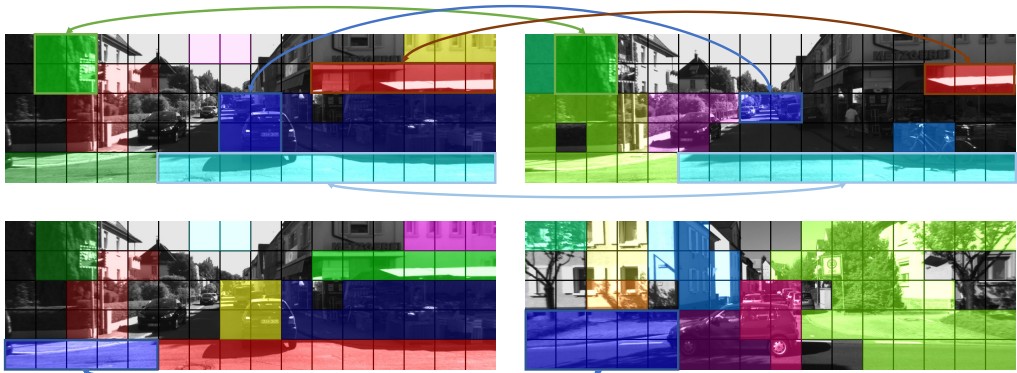

**Figure 7.** Two sets of matching images result in the KITTI07 dataset, where each local region feature is indicated by a different color, and the matching features are connected by a line. The top result is the correct match while the bottom one is incorrect.

### 4.5. General Performance

We evaluated the performance of our loop closure detection approach on four KITTI datasets with loop closures: KITTI 00, KITTI 05, KITTI 06, and KITTI 07. Table 2 presents the precision and recall of our proposed approach for the final configuration on the aforementioned sequences. If the accuracy of loop closure detection is low, it indicates a significant number of incorrect results being identified as loop closure occurrences. Constructing loop closure constraints based on these incorrect results can further deviate the optimized

motion trajectory. Conversely, a low recall rate in loop closure detection leads to the omission of many correct loop closure instances, resulting in the loss of valuable loop closure constraint information. This, in turn, may prevent the correction of the motion trajectory or weaken the effectiveness of the corrections made. The proposed method achieved high accuracy and recall rates in all four sequences, with only a few false positives observed in two of them. Following the completion of loop closure detection, additional loop closure verification can be employed to eliminate false matching results and improve accuracy. A high recall rate ensures successful detection of nearly all loop closure locations and the construction of corresponding loop closure constraints. At the same time, a high accuracy rate ensures that no false matching loop closure results adversely affect the estimation of the motion trajectory.

**Table 2.** Precision and recall of our proposed approach for the final configuration.

| Dataset | Image | Precision (%) | Recall (%) |
| --- | --- | --- | --- |
| KITTI 00 | 4007 | 93.52 | 76.53 |
| KITTI 05 | 2135 | 92.49 | 74.76 |
| KITTI 06 | 995 | 100 | 77.64 |
| KITTI 07 | 756 | 100 | 66.67 |

Figure 8 shows the evolution of the vocabulary size for these four sequences. The traditional BoW model typically uses feature points to construct a vocabulary tree, where dozens or even hundreds of feature points can be extracted from an image. This type of BoW model constructs the vocabulary tree from training sets during the offline stage. To ensure adaptability to complex and changable environments, the vocabulary size should not be too small. Consequently, these constraints result in vocabulary sizes reaching $1 \times 10^5$. For instance, the FAB-MAP 2.0 algorithm employs a vocabulary tree consisting of up to 100,000 words. Similarly, the iBoW-LCD algorithm performs online vocabulary tree construction on a large-scale sequence such as KITTI 00, which includes up to 4000 images, resulting in a final vocabulary tree containing over 800,000 words. However, our proposed approach demonstrates significantly smaller vocabulary sizes, even when applied to KITTI 00, the number of words remains below 1200, which differs by two orders of magnitude compared to traditional methods' vocabulary sizes. This reduction in vocabulary size greatly enhances the computational efficiency of the loop closure detection algorithm.

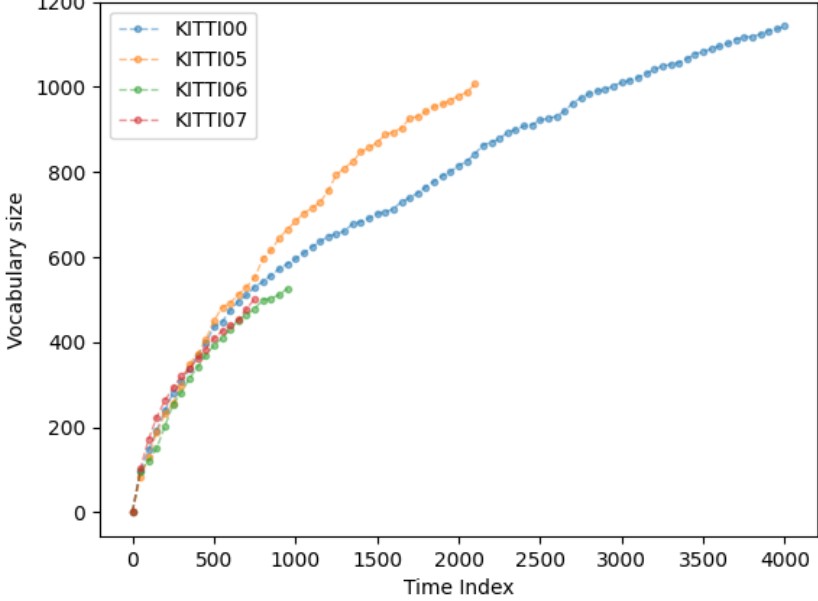

**Figure 8.** The evolution of the vocabulary size for these four sequences.

Figure 9 illustrates the response time of our proposed approach per image for these four sequences. It can be observed that, for time indices before 400, the response time remains consistently below 0.3 ms. This can be attributed to the small vocabulary size, resulting in a relatively small amount of data to process during vocabulary tree updates and image retrieval. As the number of image frames increases, the update frequency of the vocabulary tree accelerates, and the number of potential loop detection matching images increases. Particularly in scenarios where numerous images exhibit similar environments, a larger number of images necessitates calculations and comparisons, which affects the efficiency of loop detection. Nevertheless, even in such cases, the response time per image can still be maintained within 70 ms.

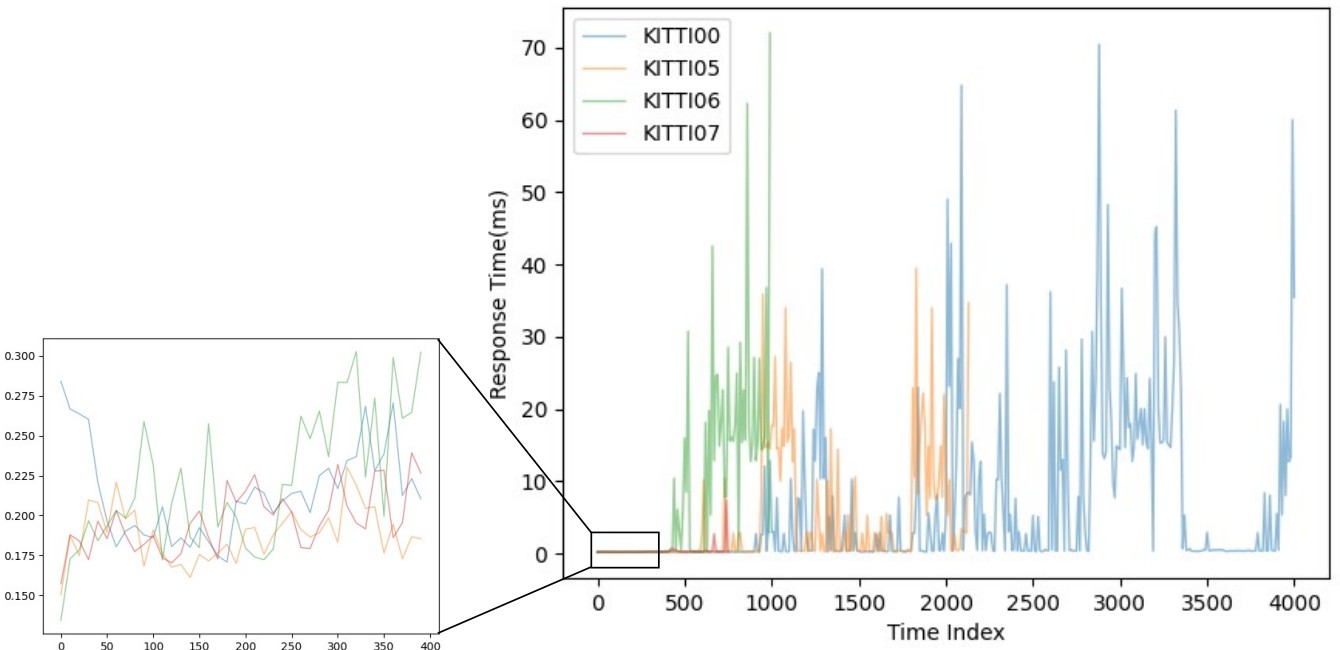

**Figure 9.** The response time of our proposed approach for these four sequences.

## 5. Conclusions

In this work, we proposed an appearance-based loop closure detection algorithm that uses an incremental Bag of Words scheme with gradient orientation histograms to retrieve similar images. The proposed local region feature extraction method extracts local region features by dividing image blocks and clustering them based on gradient orientation histograms. Our online, incremental BoW model is based on the BIRCH algorithm to generate and update a vocabulary CF tree that can maintain a small vocabulary size under large image datasets. The CF attribute of the vocabulary CF tree can estimate the frequency of word occurrence in real-time, enabling the calculation of image similarity. Finally, islands are generated by grouping highly similar images that are close in time, and loop closure candidates are obtained through competition between islands. We evaluated the proposed approach on publicly available datasets and demonstrated its ability to generate high recall and precision. A loop closure detection algorithm with high accuracy and recall plays a crucial role in achieving reliable place recognition during the operation of a SLAM system. By incorporating loop closure constraints, the cumulative error in the estimated trajectory can be effectively corrected, thus significantly enhancing trajectory estimation accuracy and overall mapping quality.

While our proposed approach demonstrates excellent accuracy and efficiency, there are still areas that warrant further improvement. For instance, we recognize the need to refine the shape restriction of extracted features by grid-divided image blocks and enhance the quality of the vocabulary tree constructed using the BIRCH algorithm. Regarding

future work, we intend to focus on local region feature extraction and vocabulary tree construction. We will investigate other local region feature extraction methods, including methods for image region selection and feature description. In addition, we plan to optimize the construction and updating processes of the vocabulary tree by implementing staged optimization and pruning approaches, which can optimize the structure of the vocabulary tree. Furthermore, our plan involves integrating our loop closure detection method into a comprehensive SLAM framework. By integrating our method into the complete SLAM system, we anticipate a substantial improvement in the overall performance and accuracy of the system.

**Author Contributions:** Conceptualization, Y.L. and W.W.; methodology, Y.L. and H.Z.; software, Y.L.; validation, Y.L., W.W. and H.Z.; formal analysis, Y.L. and W.W.; investigation, Y.L., W.W. and H.Z.; resources, Y.L. and W.W.; data curation, Y.L. and H.Z.; writing—original draft preparation, Y.L.; writing—review and editing, Y.L.; visualization, Y.L.; supervision, H.Z.; project administration, W.W. All authors have read and agreed to the published version of the manuscript.

**Funding:** This research was funded by the National Natural Science Foundation of China, grant numbers 61573148; 2020 Guangdong Higher Vocational Colleges Industry and education integration innovation platform project, project number 2020CJPT026.

**Institutional Review Board Statement:** Not applicable.

**Informed Consent Statement:** Not applicable.

**Data Availability Statement:** Not applicable.

**Conflicts of Interest:** The authors declare no conflict of interest.

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
