# Peer review of "Incremental Bag of Words with Gradient Orientation Histogram for Appearance-Based Loop Closure Detection"

_applsci, doi:10.3390/app13116481_

Round 1
Reviewer 1 Report
The authors are interested in their work to tackle the problem of image detection and feature extraction in domain of smart vehicule, more specifically loop closure detection mechanism. The present paper aims to propose an incremental bag of words with gradient orientation histogram using BRICH algorithm.
The paper well-constructed yet some part need to be carried out.
An overview of the state of the art is presented.
The contributions of the research are presented.
The paper is in its current state and is cannot be considered for publication, yet some changes need to be carried out.
• In the abstract section, the authors present an abbreviation without mentioning its meaning (BRICH and CF).
• When the authors cite a work, I recommend adding a space between the citation and text.
• The discussion section is missing in the paper. The authors should present an extended section presenting the details of the analysis of the results.
• It would be good if the author presents the motivation for the used method.
• In the references section, some works are too old (2008, 2007, ...). I recommend the authors to update some of them.
• It would be good if the authors added an additional subsection presenting a description of the used datasets.
• Since the proposed method is used in vehicles, it will be critical for the response time. I would like to see some comments about this matter.
• For the conclusion section, it would be good if the authors added some future work and showed its perspective.
Reviewer 2 Report
Regarding the content, I do not have any changes to recommend, it makes a good literary review to support the relevance of the problem to be studied and a good structuring of the content, it uses the correct methodology for this type of study and it is a consistent and well-detailed methodology to give significance to the results they show, makes a good discussion of the results with respect to the studies carried out previously, and marks the conclusion obtained well.
Although I advise looking at these things:
Respect the template provided by the journal, the first page does not have the journal logo.
In the section “5. Conclusions”, it is necessary to develop a deeper analysis of the conclusions, implications and limitations of the study. In addition to the possible future lines of research opened with this research.
And the references in the 'References' section must follow the model set by the journal. You must correct the errors that exist. Look at this in the template.
Round 2
Reviewer 1 Report
The authors have answered the questions.